# Qualitative study of GPs' views and experiences of population-based preconception expanded carrier screening in the Netherlands: bioethical perspectives

Sofia Morberg Jämterud ,[1] Anke Snoek,[2] I M van Langen,[3] Marian Verkerk,[4] Kristin Zeiler[1]

¹Department of Thematic Studies, Linköping University, Linkoping, Sweden
²Department of Health, Ethics and Society, Maastricht University, Maastricht, The Netherlands
³Department of Genetics, University of Groningen, Groningen, The Netherlands
⁴Department of Internal Medicine, University Medical Center Groningen, University of Groningen, Groningen, The Netherlands

**Correspondence to**
Dr Sofia Morberg Jämterud;
sofia.morberg.jamterud@liu.se

## ABSTRACT

**Objective** Between 2016 and 2017, a population-based preconception expanded carrier screening (PECS) test was developed in the Netherlands during a pilot study. It was subsequently made possible in mid-2018 for couples to ask to have such a PECS test from specially trained general practitioners (GPs). Research has described GPs as crucial in offering PECS tests, but little is known about the GPs' views on PECS and their experiences of providing this test. This article presents a thematic analysis of the PECS practice from the perspective of GPs and a bioethical discussion of the empirical results.

**Design** Empirical bioethics. A thematic analysis of qualitative semi-structured interviews was conducted, and is combined with an ethical/philosophical discussion.

**Setting** The Netherlands.

**Participants** 7 Dutch GPs in the Netherlands, interviewed in 2019–2020.

**Results** Two themes were identified in the thematic analysis: 'Choice and its complexity' and 'PECS as prompting existential concerns'. The empirical bioethics discussion showed that the first theme highlights that several areas coshape the complexity of choice on PECS, and the need for shared relational autonomous decision-making on these areas within the couple. The second theme highlights that it is not possible to analyse the existential issues raised by PECS solely on the level of the couple or family. A societal level must be included, since these levels affect each other. We refer to this as 'entangled existential genetics'.

**Conclusion** The empirical bioethical analysis leads us to present two practical implications. These are: (1) training of GPs who are to offer PECS should cover shared relational autonomous decision-making within the couple and (2) more attention should be given to existential issues evoked by genetic considerations, also during the education of GPs and in bioethical discussions around PECS.

## INTRODUCTION

Preconception expanded carrier screening (PECS) aims to provide prospective parents with knowledge regarding the risk of conceiving a child with a genetic

### Strengths and limitations of this study

► The qualitative design of the study gives an in-depth perspective on general practitioners (GPs) views and experiences of the practice of preconception expanded carrier screening.

► Few qualitative studies that include semistructured interviews with GPs on preconception expanded carrier screening have been published: this article, therefore, contributes to this area of research.

► Empirical bioethics as a methodological approach analyses with sensitivity the experiences of GPs of preconception expanded carrier screening in combination with a discussion of the ethical complexities and concerns related to the practice.

► The study involved seven semistructured interviews, which can seem like a small number but was deemed sufficient, since saturation was reached.

condition[1] and to enhance their reproductive autonomy.[2–6] Each child from a couple in which both partners are carriers of a mutation for the same autosomal recessive disorder runs a one in four risk of having the condition. The Netherlands has, as the first country in the world, been offering a couple-based PECS test to non-consanguineous couples who have no known genetic condition in the family. It has been offered preconceptionally by general practitioners (GPs), free of charge, as part of a pilot study, and it also included pretest counselling.[7 8] Since mid-2018, all couples in the Netherlands can ask to have a test from one of six specially trained GPs. The test is couple based, meaning that both partners are tested in parallel, and individual test results are not given. The results are presented solely as the genetic risk that a potential future child would run (carrier couple or not), as the aim of the test is to provide results that are important for reproduction.[8] This distinguishes it from many

practices with individual-based test results.[9 10] A couple for whom the result is positive can, for example, choose to be become pregnant following in vitro fertilisation combined with preimplantation genetic diagnosis.[11]

Both in the Netherlands and internationally, GPs have been described as important in offering population-based PECS.[12] Some studies have shown that the public prefers that GPs offer the test,[13 14] rather than clinical geneticists or midwives,[11] and it is probable that GPs will receive more questions about PECS as general knowledge of the practice increases.[15] However, little is known about GPs' views and experiences of PECS. Previous qualitative research has been conducted on the views and experiences of GPs related to the feasibility of the test, within an implementation study.[8] However, the current study has another focus: it examines a wider range of views and experiences of PECS and does not only focus on feasibility of the test. The aim of this article is to present an empirical bioethics analysis of the PECS practice from the perspective of GPs.

## METHODS
### Setting
In the Netherlands, the PECS test and pretest counselling was developed by van Langen and colleagues at the University Medical Centre Groningen (UMCG) in 2016, as part of a pilot study.[7 8 11 16] They have trained GPs in non-directive pretest counselling. The GPs do not perform post-test counselling but refers couples to clinical geneticists. The test is not covered by insurance for parents with normal risk, and it costs €950 per couple. This charge covers the cost of a DNA-lab test performed at UMCG. The test currently includes 70 serious early onset autosomal recessive genetic conditions. Previous carrier screening tests have often targeted fewer genetic conditions, been offered to specific high-risk groups or been offered as commercial tests outside of the primary care system.[3 17] Internationally, much genetic carrier screening is offered in early pregnancy.[18–20] However, this is not the case in this study, since the focus is only on a test that is taken before conception.

### Data collection and analysis
The study comprised seven semistructured interviews with GPs. Interviewees were recruited through the UMCG and the Northern GPs' Association in the Netherlands. A letter describing the project was sent out to GPs in which interested GPs were asked to contact the research team. Inclusion criteria were working as a GP and willing to share their professional views and reflections on the test. Both GPs who offered the screening and those who only referred couples for it were interviewed. The majority of GPs were situated in rural areas. Interviews were conducted by the second author, either in person (September 2019 to February 2020) or via phone (March to May 2020). The in-depth interviews lasted around 1 hour were recorded, transcribed verbatim and

pseudonymised. Which GP belonged to which group will not be stated in the results in order to ensure anonymity. The study examined the GPs' views and/or experiences on the practice of PECS and the interview guide covered: first impression of the test, implications of the test, experiences with patients and how the test could be improved (online supplemental file).

The qualitative semistructured design allowed interviewees to expand on issues that they saw as important. The interviews were detailed and rich in content. They offered what, in qualitative research, is called 'thick' descriptions, which allow us to consider contextual detail as part of the interpretations and analysis of meaning.[21] The aspect of saturation is essential,[22–24] the later interviews did not bring out new themes, but added to themes already present.[25 26] For this reason, the number of interviewees was deemed sufficient.

Thematic analysis of the data was conducted.[27] AS, SMJ and KZ read all interviews independently of each other and carried out an initial coding. AS, SMJ and KZ carried out independent coding of the data, independently identified subthemes based on this coding and jointly clustered subthemes into broader patterns of meaning, that is, themes. NVivo, software designed to analyse qualitative data, was used. This process, guided by the aim of the study, can be described as the researchers engaging with and interrogating the data, back and forth, and developing themes. The Standards for Reporting Qualitative Research (SRQR) has been followed.[28]

### Ethics approval statement
The interviewees were informed by letter and orally about the project. Participation was voluntary and participants gave written, informed consent prior to the interview. This article is the result of a Dutch-Swedish collaboration to examine patients' views and experiences of PECS and GPs' work experiences and views of PECS.

### Empirical bioethics
The study's methodological framework is empirical bioethics,[29–31] a growing field of research.[32] Empirical bioethics is a heterogeneous field that combines empirical research—commonly qualitative empirical research—with an ethical or philosophical analysis.[31 33] Just as other qualitative research methods, it involves a detailed analysis of descriptions and views given by interviewees on a particular subject and a focus on complexities. However, the particular value of empirical bioethics rests with the way the qualitative analysis is combined with, for example, conceptual analysis and philosophical and ethical discussion.[29 31 33] The combination of qualitative analyses with philosophical or ethical analyses has proven to be of much value: it can refine an ethical discussion within a medical practice through its close attention to concerns that arise within this practice, without losing sight of the specific context, while ensuring that theoretical philosophical and/or ethical discussions contribute to concerns within the concrete medical practice. In this way, such combined

analyses can contribute to the improvement of care. In the present study, we identify themes that include concerns held by the interviewees, engage with the results of the thematic analysis, identify norms and values, contextualise the identified themes against previous relevant analyses and discuss the empirical findings in relation to previously identified ethical concerns and discussions (here called an empirical bioethical discussion).

## Patient and public involvement

No patients or members of the public were involved in the study.

## RESULTS

The first theme identified in the thematic analysis of interviews with GPs on PECS is 'choice and its complexity'. After presenting the thematic analysis, we offer an empirical bioethics discussion and argue that it highlights the need for facilitating shared relational autonomous decision-making within the couple. The second theme is 'PECS as prompting existential concerns', which includes two subthemes: 'prevention of suffering' and 'the test within the framework of societal concerns'. We also discuss this theme in the context of bioethics and argue that it should preferably be understood in terms of an entangled existential genetics that brings out ethically pertinent aspects of the practice of PECS.

## Choice and its complexity

GPs stated that they valued PECS because it increases reproductive choice. All stated that non-directiveness was an important ethical condition when offering the screening, underlining the importance of providing adequate information, so that the couple could make an informed choice: 'We don't force our opinion of matters, but discuss the possibilities and then the choice is theirs' (R1). Some GPs were concerned about whether couples understood what the test tests for and pointed to difficulties in obtaining an overview and understanding of the conditions that are screened for, particularly since some of the conditions are very rare. They also wondered whether couples understood what a positive test result implies, namely, that they might have to make decisions on reproduction and/or their relationship. GPs were concerned about how the difficulty of grasping such implications affected the couples' informed choice. GPs described that giving adequate information is an important part of explaining the various possible results. Couples needed 'at least an understanding that, if something came out [they tested positive], there would be consequences' (R7), and that these consequences may affect the couple's relationship. The GPs were also concerned about the effect of a positive test result on couples and their personal identity, stating that when discussing the range of conditions that were tested for it became clear that 'it can also be a huge burden, knowing what genes you carry' (R7).

GPs held the view that attention should be given to the views and reflections of potential parents about what may happen after a positive test result, and more broadly on their views and vision of life:

> You shouldn't educate people to frighten them, that's not the point, but you have to be honest. However, you also touch—you also touch a philosophical aspect, a spiritual aspect. How do you feel about life? It's just not the same for everyone. A lot of people have some kind of idea about that. They haven't thought about it very deeply, but they do have an idea. For some people, it [the test] just doesn't fit in with their philosophy of life, you know. For a lot of people, I think. (R7)

GPs described how a complex situation arose when couples had different views on the test and struggled with shared decision-making as a couple. One GP described how one couple could not agree on what to do after they had tested positive, and later found out that the couple had split up before they were informed about the test results. Another GP recalled how a couple could not agree on whether to take the test:

> Both partners should agree on taking the test. And to one couple I said: "Maybe you should go home and think about it, and you can always decide whether to take the test or not. And if you want to come again and talk about it, you are welcome". […] If they can't agree, they should not do it. Because then if you test positive, you have a problem. (R5)

This quotation also illustrates the potential complexity of making this choice together, as a couple.

## Complexity of choice and shared relational autonomous decision-making: an empirical bioethics discussion

The theme of choice described above resonates with a common motivation and argument for PECS that revolves around reproductive autonomy, underlining that parents should be informed about the risks of giving birth to an affected child. Based on this information, one can make an informed choice.[8] However, the results of the thematic analysis point to four areas that coshape the complexity of choice in relation to PECS, from the perspective of GPs. These areas are presented separately, but are closely intertwined: (1) medical aspects—the couples must understand what is being tested: the medical conditions may be rare and it may be difficult for them to get clear grasp of the what the conditions involve, (2) moral aspects—this area concerns the moral values and ideals held by the couple, and how they are expressed in the decision of how to act if the result is positive. The choice of whether to undertake PECS must also be related to the couples' views and visions of life, (3) social aspects. The implications that this test may have for the couples' relationship, and how personal choice relates to societal structures, (4) psychological aspects. The GPs pointed to a concern for

the burden that may be experienced in knowing that one is a carrier.

Since the test is couple based, the fact that the decision is shared between the partners is underlined. As noted, respect for autonomy is often a key concern in PECS.[2–6] However, while previous discussions of autonomy have sometimes been based on an individualistic interpretation of autonomy, many have argued that relational aspects of autonomy must be considered.[34 35] It is pointed out that this is a process in which an independent, self-sufficient patient who reaches a decision without interaction with others is not consistent with the relational character of many forms of healthcare.[36 37] Conceptions of relational autonomy show how social relationships are significant in developing and exercising autonomy: our very ability to make choices is shaped in and by the social context in which we live, and this ability is constrained by the same context.[34] Relational aspects of autonomy can be unpacked using the concept of shared relational autonomous decision-making, which in this case will be shared decision-making *between the partners*. This specifically addresses conditions for decision-making being *shared* and draws attention to that which takes place between partners who need to come to a shared decision about PECS. Shared relational autonomous decision-making may, therefore, require that both partners have the ability and opportunity to reflect on their vision of life as a family: what matters to each of them individually, what matters to the other partner and what matters to them as a couple. They may also need to have the ability and opportunity to engage in a decision-making process together, and they must reach a shared decision that both find acceptable.[38]

Shared relational autonomous decision-making can, therefore, be a complex process involving several aspects such as medical, moral, social and psychological aspects (see figure 1). A broader discussion of the role of families in healthcare is pertinent to this discussion.[39]

### PECS as prompting existential concerns

The second theme was centred on how PECS prompts, produces or can be seen as a response to existential concerns at the levels of the couple/family and of society. This theme consists of two subthemes: 'prevention of suffering' and 'the test within the framework of societal concerns'.

### Prevention of suffering

This first subtheme was centred on suffering as an existential issue related to PECS. The GPs reflected on the PECS test as a means to prevent possible physical and emotional suffering on the part of the families and the potential future child. Some emphasised the prevention of suffering as a reason for offering the test. One GP stated: 'Then you first have to explain […] what you want to prevent through carrier screening', namely 'the prevention of much suffering' (R2). The nature of suffering was multilayered, where prevention of physical suffering was one layer. One GP described what one of the genetic conditions that the test can identify (epidermolysis bullosa) can mean for a child:

> …the blister condition […] exists in various forms […] it's a very short life expectancy. The babies, at the moment you touch them, a blister forms. So that means that those children will acquire a burn on

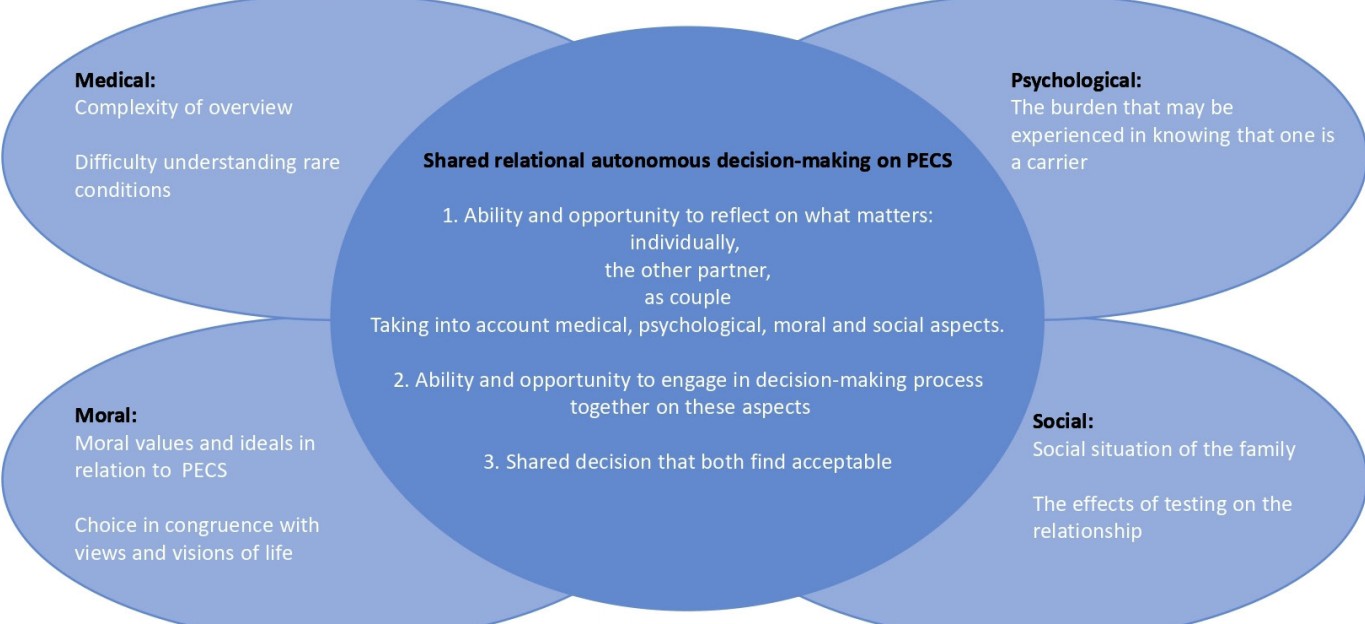

**Figure 1** The empirical bioethics discussion of the theme "choice and its complexity" showed that several areas—medical, psychological, moral, social—co-shape the complexity of choice on PECS. All these areas need to be reflected on, not only by each individual but also by the couple, together. Hence the couple need to engage in a shared relational autonomous decision-making on these areas.

more than ninety per cent of their body surface area in practically no time. […]. (R5)

GPs also described the suffering of the family as a whole, stating that living with a severely ill child is not only emotionally difficult but also brings relational, financial and work-related concerns that can cause great distress:

If you've been in my profession for a while, you see the extremely disastrous consequences of a new family losing their child to disease. Parents who often experience the full bedside, give up their job, and subsequently never mentally get back on track. (R1)

Prevention of suffering was presented as a concern relating to the whole family and as potentially existentially devastating. This way of describing suffering helped to position PECS as a response from the medical care system to a devastating situation.

### The test within the framework of societal concerns

The second subtheme was centred on the feeling of responsibility to society, the risk of a less inclusive society and blame as an existential issue related to PECS.

GPs reflected on what the opportunity to take a screening test could mean for parents' feelings of responsibility to society, if/when giving birth to a child with a condition that could have been screened for. One GP stated: 'I think the parents just have the responsibility of being able to prevent having a handicapped child. That burden, of course, is primarily borne by society as a whole. […] I think the general public has to have an opinion about that, whether you wish to bear these things as a society' (R1). 'Balancing' the costs to society of offering a genetic test against the costs to society of providing care for a child with a severe disease was described:

New ways are continuously being found to perform genetic tests in a simpler manner, and thus more affordable. And, as it grows more affordable… there's always a kind of balancing act, between how expensive a test is, and how expensive it is to have such a child. That's a fairly cynical consideration, of course. (R5)

Furthermore, GPs were concerned about the possibility that screening may result in a less inclusive society:

[…] you can test to prevent people from getting ill, or that ill people are born. But, at the same time […] you want a more inclusive society for everyone […]. That you shouldn't judge someone if they don't want to take such a test, despite you offering it. (R5)

Some GPs expressed concerns that parents who give birth to a child with a condition that the test could have identified would be blamed for conceiving without having taken the test:

To what extent should you tell people that they have a risk and that they can prevent this? Because at a certain moment, the situation will arise that there are people who have a child with an illness and they could have prevented this. And this is accompanied by blame and penitence, right? (R2)

### Entangled existential genetics: an empirical bioethical discussion

Values and norms that are central in this theme are the value attributed to preventing suffering, norms pertaining to responsibility for knowledge about genetic risks and about preventive measures after obtaining such knowledge and concerns with normative shifts in society, that is, shifts in the understanding of what one should do when considering whether to try to become pregnant. In the following, we show that the subthemes, when brought into dialogue with each other, underline an intricate interplay, an entanglement, in which the existential issues raised by PECS rest on *both* the level of the couple/family level *and* on the level of society and that these two levels influence each other. They are entangled. The concept of entanglement is commonly used to describe a situation in which parts that at first glance are regarded as being separated from one another are actually inseparable and thoroughly intertwined.[40] Concerns on the two levels (couple/family and society) inform each other—and the former cannot be fully understood without the latter. As one such example, research has shown that couples' choices are shaped by social values and norms, which differ between sociocultural contexts. This means that choices, even if autonomous, are influenced by the context in which one lives, and this context not only makes certain discussions and choices possible but also sets limits on them.[41] As another example, the view of PECS as a method to prevent suffering needs to be understood against other concerns and ideas, such as the idea that to be responsible, an act must consider societal concerns. When these views are understood relative to each other, they help explain how PECS choices made by couples can come to be positioned as concerns both on the couple/family level and on the societal level—and these levels are entangled. As yet another example, previous research has examined whether PECS results in, or strengthens, the idea that parents have a responsibility to prevent giving birth to a child with a severe condition or disability.[42] A similar concern was raised in the GP interviews, namely, that parents may be perceived as having a responsibility to use screening. The notion of entanglement helps to demonstrate the intertwinement of these layers of concern. If a couple understands genetic responsibility to be a societal norm and value, this understanding can influence their choice. Furthermore, couples' choices can, in turn, result in shifts in societal norms towards less inclusive societies. An analysis in which existential concerns related to PECS are seen as an entanglement between the couple/familial and societal levels can shed light on the family-society dynamic. For this reason, what needs to be focused on is what here has been labelled 'entangled existential genetics', existential

dimensions that are prompted or evoked by medical practices such as PECS, and in which the interests of the couple/family and society are entangled.

## DISCUSSION

We have shown in this study the need to acknowledge the complexity of choice in relation to PECS, pointing to the co-shaping of medical, moral, social and psychological areas of concern, and the importance of facilitating shared relational autonomous decision-making within the couple in counselling on PECS. Furthermore, we have discussed 'entangled existential genetics'. This idea emphasises that the existential issues raised by PECS rest *both* on the level of the couple/family level *and* on the level of society. These two levels influence and inform each other—they are entangled—and the former cannot be fully understood without the latter. Normative shifts in society can have an impact on what couples come to perceive as the choices available to them, and this in turn can have consequences for societal views on PECS.

A strength in our study is the methodological approach of empirical bioethics. It allows us to combine qualitative analysis with an ethical discussion, maintaining a detailed analysis of descriptions of PECS by GPs, a group of clinicians who may play a crucial role in offering PECS[8 12–14] and an analysis of ethical complexities. The in-depth nature of the interviews is also a strength of this study even though the relatively small sample in the interviews could be seen as a limitation. However, we concluded that saturation had been reached, since later interviews did not bring out any new themes. This study is based on the practice of PECS as developed within a pilot study in the northern parts of the Netherlands. This means that it may be difficult to generalise our conclusions to other contexts, especially since the test has been developed as a couple-based test. Our results, however, show several aspects that may be interesting also in the context of individual-based tests, such as the identity of the areas that co-shape the complexity of choice.

Previous research has discussed the broad range of conditions that can be screened for in PECS and how this relates to the complexity of consent.[3] Ethical concerns have been voiced about the difficulty of obtaining an overview of the conditions that are screened for, and what these conditions can mean. It has been discussed whether this difficulty hampers or has other negative effects on the patients' informed decision-making.[43] The GPs interviewed acknowledge such complexity of overview. However, our results point to a broader complexity in relation to choice, not limited to obtaining an overview, but where the co-shaping of different areas adds to the complexity of choice. Handling such complexity may add to the difficulty for GPs when providing counselling on PECS. Previous research has pointed to the need for education of physicians and GPs if they are to play the role of counsellors on PECS,[8 44] in particular, for the need for an awareness of the genetic competence of GPs when

counselling on PECS.[15] However, previous studies have acknowledged to a lesser extent the *complexity of choice* that needs to be acknowledged in counselling on PECS.

We offer two recommendations for future GP practices and bioethical discussions of PECS:

► Careful training in non-directive genetic counselling of GPs should cover *shared relational autonomous decision-making within the couple*. Couples should reach a decision following a discussion within a healthcare setting, with healthcare professionals who have been tasked with facilitating the decision-making. Since couple-screening specifically focuses on the couple, and because partners may have different views and values that affect the decision of whether to undergo screening, a more specific training in how to promote shared relational autonomous decision-making within the couple is called for.

► Attention should be given to what we term *entangled existential genetics*, that is, existential dimensions prompted or evoked by PECS in which the two levels of couple/family and society are entangled. In counselling, this can mean that existential dimensions concerning the meaning of the testing for self-understanding are addressed, and the counselling could include how to live with a positive result. Furthermore, entangled existential genetics highlights the importance of political and socioethical reflections on existential concerns in bioethical discussions on PECS.

**Acknowledgements** The authors would like to thank the participating GPs and Northern GPs' Association AHON (Nynke Schouwenaar/Dorina van der Kolk (PA)).

**Contributors** All authors approved the content of the manuscript (SMJ guarantor). AS collected the data, while AS, SMJ and KZ conducted the thematic analysis. SMJ and KZ took primary responsibility for writing the section of the manuscript that deals with the empirical bioethics results, and the discussion. IMvL contributed medical expertise and ethical perspectives on ECS practice. MV contributed a family ethics perspective, and SMJ, AS, MV and KZ contributed medical ethics expertise.

**Funding** This research is part of the research programme 'A Feminist Approach to Medical Screening' (grant number 2016-00784), which has received funding from the Swedish Research Council (Vetenskapsrådet). We thank the Swedish Research Council for its funding of the research.

**Competing interests** None declared.

**Patient consent for publication** Not applicable.

**Ethics approval** The substudy with patients was submitted to and approved by the Medical Ethics Review Board of the UMCG, Groningen (reference number 2019/355), which stated that this study on GPs was not subjected to the Dutch Medical Research Involving Human Subjects Act. The substudy with patients was also submitted to and approved by the Swedish Research Ethics Board (reference number. 2019-04501), while the substudy with healthcare personnel was exempt from review, since it focused on their views and experience of their work, in a way that did not address sensitive personal data (personal communication with the scientific secretary of the board, 31 October 2017).

**Provenance and peer review** Not commissioned; externally peer reviewed.

**Data availability statement** No data are available.

of the translations (including but not limited to local regulations, clinical guidelines, terminology, drug names and drug dosages), and is not responsible for any error and/or omissions arising from translation and adaptation or otherwise.

**ORCID iD**
Sofia Morberg Jämterud http://orcid.org/0000-0003-2998-3971

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
