## [Reviewer comments · BMJ Open]

ARTICLE DETAILS

TITLE (PROVISIONAL)	A qualitative study of GPs' views and experiences of population-based preconception expanded carrier screening in the Netherlands: Bioethical perspectives
AUTHORS	Morberg Jämterud, Sofia; Snoek, Anke; van Langen, I.M.; Verkerk, Marian; Zeiler, Kristin

VERSION 1 – REVIEW

REVIEWER	Matar, Amal Uppsala Universitet
REVIEW RETURNED	04-Sep-2020

GENERAL COMMENTS	The authors have a unique angle since they were able to ask GPs who have offered and presented the ECS. This gives the discussion on ECS a much needed perspective because as the paper presented, the GPs were acting as counselors as well. Methodology: Having said that I think the number of interviews are small to reach saturation of results which the authors failed to mention. This is reflected in the number and quality of quotes. The analysis/interpretations that follow are not necessarily conducive to the quotes. I have requested for major revisions, the need for more interviews till the authors reach a saturation of data and indicate so in their methodology. Results: The bio-ethical analysis sections under both major themes are better if relocated under the discussion section which can end with the recommendations the authors propose, since the analyses are more about interpreting the results and relating them to relevant literature. Research Ethics: There is no mention of an ethical review of the study. Do interviews with GP (healthcare professionals) in the Netherlands require no ethical review? If so can the authors refer to the law or regulations that state so? Conflict of interest: One of the authors designed the preconception screening test, but i am not sure if that means IP or royalties and according to authors' description that test is not reimbursed for potential parents if they order it - i am not sure if that means that potential parents have to buy it themselves privately and the test is not covered by the healthcare system or health insurance company?
---

	Bio-ethical analyses: Lastly, there are vague descriptions that need more clarification under the second theme, the concepts of entanglement and existential genetics. They are very interesting concepts but need further clarification in relation to the results presented. Looking forward to reading the reviewed version. - The reviewer provided a marked copy with additional comments. Please contact the publisher for full details.
--	---

REVIEWER	Van der Hout, Sanne Maastricht University
REVIEW RETURNED	08-Sep-2020

GENERAL COMMENTS	This manuscript has a lot of potential. However, the method and overall structure of the text remain somewhat unclear. In order to make this manuscript suitable for publication, I would recommend a number of clarifications, additions and adjustments. Introduction  - The first sentence is very long and therefore, it is not immediately clear that the Netherlands is the first country in the world enabling an ECS offer to the general population by GPs. - Why does the UMCG use a couple-based approach? - How much do participants need to pay for taking part in ECS? - You mention that GPs were preferred as providers of the test. Preferred over whom? Clinical geneticists, midwives, commercial providers, ...? - Considering that 'opportunities of prevention' is a recurring theme in the Results section, it is important to explain already in the introduction what means of prevention are enabled by preconception ECS (PGT-M, gamete donation, etc.). Method and structure  - It is not entirely clear how the two main themes 'Choice and its complexity' and 'Preconception ECS as prompting existential concerns' were identified. Moreover, these themes are quite broad and remain somewhat abstract. What was your motivation for selecting two broad themes instead of a greater number of more elucidating subthemes? - 'Empirical bioethics' as part of your method requires further clarification. What exactly is the purpose of using this method? What is the added value of this method compared to other methods (used in ethics as well as in the social sciences)? Moreover, your explanation that empirical bioethics combines empirical research with a philosophical or ethical analysis needs to be clarified. What kind of ethical or philosophical analysis are you referring to? Is your approach normative, phenomenological, analytical, ...? - You mention that 'The empirical bioethics method sometimes leads to practical recommendations.' This formulation does not really speak in favor of using this method. How to ensure that the empirical bioethics method can be of practical value? And if it does not lead to practical recommendations, does it have any other value? Data collection and analysis  - I think it is important to specify whom of the GPs offered ECS and whom merely referred couples to colleagues. Was there
--

	a specific reason for also selecting members from the latter group? Were those who had more experience with offering ECS more positive or more critical towards this screening offer? Results  - Your Results section would become more structured if you would make use of (more) subthemes. E.g. the theme 'Choice and its complexity' could be divided into subthemes such as 'Informed consent', 'Non-directiveness', 'Pre-test counselling', etc. Moreover, the subtheme 'Implications of the test' is very broad. It might be helpful to draw a distinction between 'personal' and 'societal' implications of the test. - The notions of 'generic consent' and 'shared decision making' need to be explained more thoroughly. Discussion  - In this section, you could elaborate more on the way in which preconception ECS leads to complicated questions with regard to the responsibility of carrier couples towards their (future) offspring as well as with regard to society. This is one of the key topics in your Results section, but remains somewhat 'underexposed' in your Discussion. References  - Sometimes the titles of journals are abbreviated (e.g. refs. 1, 2 and 6), and sometimes not (e.g. refs. 3 and 5). - In ref. 6, the name of one of the authors is spelled incorrectly: Schuurmans. Language  - The standard of English is not always sufficient. Notably, plural and singular are mixed up many times. E.g., p. 5, 51/52: "GPs has also been described as..."; p. 6, 6-9: "Empirical bioethics is a heterogeneous field that combine empirical research with..." - P. 9, quotation at the end of the page: "... and it is not the idea that they get in such a fight that they child won't even come into existence..." - American and British English are intermixed.
--	---

VERSION 1 – AUTHOR RESPONSE

Response to Reviewer 1. Dr. Amal Matar

We are very grateful for the careful reading of the article and the reviewer's valuable comments in order to improve the article.

Response to Dr Matars general comments:

Methodology: Having said that I think the number of interviews are small to reach saturation of results which the authors failed to mention. This is reflected in the number and quality of quotes. The analysis/interpretations that follow are not necessarily conducive to the quotes. I have requested for major revisions, the need for more interviews till the authors reach a saturation of data and indicate so in their methodology.

The reviewer has pointed to the number of interviewees and the question of saturation and mentioned that a discussion on saturation is not indicated in the methodology section. We thank Dr Matar for this

important comment and have expanded the section regarding this matter in the methods section as well as in strengths and limitations. However, concerning the view on saturation and how many interviews that are needed in order to reach saturation we differ in opinion. The discussion on saturation is ongoing within qualitative research and there are of course different views on this subject. We would argue that the interviews offered, thick descriptions, and gave an in-depth perspective on the practice and complexities of PECS. Furthermore, the latter interviews did not bring out new themes but added to the themes already present in the previous interviews, saturation was reached.

Results: The bio-ethical analysis sections under both major themes are better if relocated under the discussion section which can end with the recommendations the authors propose, since the analyses are more about interpreting the results and relating them to relevant literature.

The reviewer wished that the bio-ethical analysis section was relocated under the discussion section. Regarding the empirical bioethical analysis: We understand this as a combination of an empirical and an ethical analysis. We have therefore not wanted to break up or split the analysis in first a thematic analysis under “Results” and a bioethical analysis under “Discussion”. The fact that the analysis is *combined* is one of the factors that contributes to specific empirical bioethical approach and creates a text where the empirical and ethical discussion is kept tightly together. However, if the editor and reviewers’ still find this important we would of course consider this option.

Research Ethics: There is no mention of an ethical review of the study. Do interviews with GP (healthcare professionals) in the Netherlands require no ethical review? If so can the authors refer to the law or regulations that state so?

Please see the above statement to the editor. A statement is now included in the methods section.

Conflict of interest: One of the authors designed the preconception screening test, but i am not sure if that means IP or royalties and according to authors' description that test is not reimbursed for potential parents if they order it - I am not sure if that means that potential parents have to buy it themselves privately and the test is not covered by the healthcare system or health insurance company?

This is an important remark and of course of importance in relation to transparency. If the editor finds it important we would be happy to include a statement in the competing interest section stating that: *One of the authors (lvL) designed the preconception carrier screening test in the Netherlands but does not have any royalties from the test.*

We have also in the article described what the test now costs and that it is a cost that covers lab-costs (page 5).

Bio-ethical analyses: Lastly, there are vague descriptions that need more clarification under the second theme, the concepts of entanglement and existential genetics. They are very interesting concepts but need further clarification in relation to the results presented.

We thank Dr Matar for this important comment and have expanded what we mean by “entangled existential genetics”. We exemplify this to a greater extent in order to show more concretely what is meant and hope that these changes will prove satisfying.

Response to Dr Amal Matar’s comments in the separate document. Please see the attached file (“File nr 3”) for the reviewer’s comments.

- P. 5. Non-reimbursed – excluded the word and explained this instead.
- P. 6. The reviewer has asked for clarification on saturation and a need for more explanation on how the thematic analysis was conducted. Please see the above answers to Dr Matar for answers to the question on saturation. Regarding the thematic analysis we have added clarifying comments in the section on “Method”.
- P.7. The reviewer has asked about the need for REC. Please see above for answer to Dr Matar’s general comments.
- P. 8. Further explanations on “impact on couples” receiving positive results is asked for. Valuable comment and this part has been rewritten and we explain what this means rather than merely use the word impact.
- P 9. The reviewer asks about the meaning of thorough consequences. The text has been expanded and now explain in more detail.
- P. 9. The number of the respondent is added. Thank you for pointing this out.
- P. 10. The discussion on generic consent is omitted.
- P. 12. The theme of “prevention of suffering” has been changed in order to provide a clearer text for the reader. The part of resisting is omitted.
- P. 12: We agree with the reviewer that the quote can be understood as not being non-directive. However, this is a quote from one of the GPs and we do not comment the interviewees answers or make comments in the text about who interviewees ought or not ought to act but as qualitative researchers we describe and analyse these interviews.
- P. 13. We have changed the language to more objective language and pointed to that this is an explanation from the GPs.
- P. 15. The reviewer asks for more details on entanglement and rephrasing in order to provide a clearer analysis. This is an important comment and we appreciate it. We have added further explanations as well as added more examples of what we mean by entanglement. We hope that this is sufficient and make the analysis clearer.
- P. 16. The sentence referred to has been omitted.
- P. 17. It was important for us to see that the wording and explanation on existential genetics called for more clarification. As stated we have strived for a clearer description.

Response to Dr Sanna van der Houts’ comments.

We are very grateful for the careful reading of the article and the reviewer’s valuable comments in order to improve the article.

Please see the attached file (File nr 2) for the reviewer’s comments. However, they are also included here in bold in order to make this more readable.

Introduction

The first sentence is very long and therefore, it is not immediately clear that the Netherlands is the first country in the world enabling an ECS offer to the general population by GPs.

Important remark about the first sentence and we have changed the introduction of the article in order not to confuse the reader.

Why does the UMCG use a couple-based approach?

We have included an explanation to why UMCG choose a couple-based approach.

How much do participants need to pay for taking part in ECS?

We have included the cost for PECS.

You mention that GPs were preferred as providers of the test. Preferred over whom? Clinical geneticists, midwives, commercial providers, ...?

We have clarified the results from the research claiming that GPs were preferred as providers of the test over for example clinical geneticists or midwives.

Considering that ‘opportunities of prevention’ is a recurring theme in the Results section, it is important to explain already in the introduction what means of prevention are enabled by preconception ECS (PGT-M, gamete donation, etc.).

We have included examples of what means of prevention that are enabled by PECS.

Method and structure

It is not entirely clear how the two main themes ‘Choice and its complexity’ and ‘Preconception ECS as prompting existential concerns’ were identified. Moreover, these themes are quite broad and remain somewhat abstract. What was your motivation for selecting two broad themes instead of a greater number of more elucidating subthemes?

The reviewer wishes us to explain further how the two main themes “choice and its complexity” and “preconception ECS as prompting existential concerns” were identified. We appreciate this comment and have added clarifying comments in the article concerning the thematic analysis in the section “Method” (see below). Furthermore, the reviewer understands the themes as somewhat abstract. We appreciate this comment and have made these more concrete and also added an illustration in order to clarify our results and analysis.

“Thematic analysis of the data was conducted.[25] AS, SMJ and KZ read all interviews independently of each other, and carried out an initial coding. AS, SMJ and KZ carried out independent coding of the data, independently identified sub-themes based on this coding, and jointly clustered sub-themes into broader patterns of meaning, i.e. themes. NVivo, software designed to analyze qualitative data, was used. This process, guided by the aim of the study, can be described as the researchers engaging with and interrogating the data, back and forth, and developing themes. The SRQR reporting guidelines have been followed.[26]“

‘Empirical bioethics’ as part of your method requires further clarification. What exactly is the purpose of using this method? What is the added value of this method compared to other methods (used in ethics as well as in the social sciences)? Moreover, your explanation that empirical bioethics combines empirical research with a philosophical or ethical analysis needs to be clarified. What kind of ethical or philosophical analysis are you referring to? Is your approach normative, phenomenological, analytical, ...?

The reviewer understands that ‘empirical bioethics’ needs further clarification. We appreciate this comment and see the need for clarification of our perspective. We have therefore elaborated the text and more specifically state which consequences such a methodology have in the study.

You mention that ‘The empirical bioethics method sometimes leads to practical recommendations.’ This formulation does not really speak in favor of using this method. How to ensure that the empirical bioethics method can be of practical value? And if it does not lead to practical recommendations, does it have any other value?

The reviewer has questioned the value of the choice of method if it cannot be of practical value. We appreciate this comment and when pointed out it was clear that our statement was confusing. This

has now been changed. The also hope that the practical recommendations that are stated can be of value for GPs.

Data collection and analysis

I think it is important to specify whom of the GPs offered ECS and whom merely referred couples to colleagues. Was there a specific reason for also selecting members from the latter group? Were those who had more experience with offering ECS more positive or more critical towards this screening offer?

The reviewer would prefer that we distinguished between the GP's who offered PECS and those that did not. Even though we recognise why such a clarification could be of value to the reader we consider it problematic in relation to research ethical considerations. If the GP's who offer PECS would be pointed out they could easily be traced which we could not risk since they are promised anonymity. To include both GPs who offer PECS as well as those who do not offer was considered important in order to gain a broader material, greater complexity and more nuances within the descriptions of the practice. Those who offer PECS are positive to the practice which also is in line with their choice to offer it: However, they also describe complexities.

Results

Your Results section would become more structured if you would make use of (more) subthemes. E.g. the theme 'Choice and its complexity' could be divided into subthemes such as 'Informed consent', 'Non-directiveness', 'Pre-test counselling', etc. Moreover, the subtheme 'Implications of the test' is very broad. It might be helpful to draw a distinction between 'personal' and 'societal' implications of the test.

The reviewer has asked for more subthemes in order to make the text more structured. As it is at the moment the results are divided in two themes where one theme has two subthemes. However, as previously mentioned we appreciate that the reviewer needs more clarity in the structure and have therefore for example included illustrations in order to clarify the analysis.

The reviewer has also pointed to that the subtheme "implications of the test" was very broad pointing to that a distinction could be drawn between personal and societal implications. This was a very good point and we have decided to rewrite this subtheme now titled "The test within the framework of societal concerns".

The notions of 'generic consent' and 'shared decision making' need to be explained more thoroughly.

The notion of generic consent has been omitted and shared autonomous decision making has been explained in more detail.

Discussion

In this section, you could elaborate more on the way in which preconception ECS leads to complicated questions with regard to the responsibility of carrier couples towards their (future) offspring as well as with regard to society. This is one of the key topics in your Results section, but remains somewhat 'underexposed' in your Discussion.

This is a very important remark and responsibility is the topic of another article we are writing at the moment where we are elaborating on this question in more depth. However, in this article we want to point to the complexity of choice but if the editor would like us to expand on this topic in the Discussion we would of course do so.

References

Sometimes the titles of journals are abbreviated (e.g. refs. 1, 2 and 6), and sometimes not (e.g. refs. 3 and 5).

The references have been corrected

In ref. 6, the name of one of the authors is spelled incorrectly: Schuurmans.

The name Schuurmans is corrected.

Language

The standard of English is not always sufficient. Notably, plural and singular are mixed up many times. E.g., p. 5, 51/52: “GPs has also been described as...”; p. 6, 6-9: “Empirical bioethics is a heterogeneous field that combine empirical research with...”

The article has been sent to a professional English language reviewer.

P. 9, quotation at the end of the page: “... and it is not the idea that they get in such a fight that they child won’t even come into existence...”

Previous errors have been corrected.

American and British English are intermixed.

Since the text has been sent to a professional language reviewer this problem is now solved.

VERSION 2 – REVIEW

REVIEWER	Matar, Amal Uppsala Universitet
REVIEW RETURNED	29-Sep-2021

GENERAL COMMENTS	Thank you for your feedback and thorough responses.
---

REVIEWER	Best, Stephanie Australian Institute of Health Innovation, Centre for Healthcare Resilience and Implementation Science
REVIEW RETURNED	01-Oct-2021

GENERAL COMMENTS	A qualitative study of GPs’ views and experiences of population-based preconception expanded carrier screening in the Netherlands: Bioethical perspectives Thank you for the opportunity to review this interesting and timely manuscript. Genetic carrier screening is of interest internationally and the focus on this topic is growing. The authors use a qualitative approach adopting in depth interviews with GPs to form the basis of their bioethical discussion. While the findings were compelling, I have a few comments. The readership of BMJ Open are not all ethicists and I suspect would be interested in the empirical bioethical approach. However, I felt there needed to be further information about this approach early on in the manuscript to help guide the reader. I was unclear of the benefits (and limitations) of this analysis.
---

	Internationally, much genetic carrier screening is offered in early pregnancy so this study will have missed this. In itself this is not a problem as this study focuses on PECS but there would be value in raising testing in early pregnancy to put this study in context. In the introduction there is a comment that “Important qualitative research has been conducted...” could you expand on what this is and how is it important? I wondered if it was more that it was highly relevant to this study? The introduction closes with a sentence on what was done i.e., thematic analysis and a bioethical discussion. A clear aim of the study would be of help here, so the reader understands the purpose of the paper. In the methods section I was unclear if we needed the detail about the pilot study. I found it confused this study with (I am assuming) the previous pilot work. There were no dates as to when these activities happened. The explanation about the PECS test was helpful – while the cost covers the DNA-lab test it may be interesting to readers to outline what is not covered. Are the trained GPs expected to do pre and post test counselling? Could the interview guide be added to the supplementary information? The results section opened directly with themes and quotes from the participants. Could there be some information about the characteristics of the participants e.g metro/rural, experience with PECS as is relevant. An outline of the themes to be presented would also be of help to signpost the reader. I found the presentation of the themes a little confusing. The second theme was clearly presented though there were subdivisions that were then not followed up with more detail. I was unsure why the first theme 'choice and its complexities' is not contained with the second them of 'complexity of choice...'. The discussion about entangled existential genetics was of interest but felt like it belonged in the discussion section. Mention is made in the limitations of the relatively small sample. I would mention the in-depth nature of the interviews as a strength of this study. I hope these comments help strengthen your work. Good luck.
--	--

VERSION 2 – AUTHOR RESPONSE

Reviewer: 1

Dr. Amal Matar, Uppsala Universitet

Comments to the Author:

Dear authors. Thank you for your feedback and thorough responses.

Response: We thank Dr Amal Matar for very valuable comments.

Reviewer: 2.

Dr. Stephanie Best, Australian Institute of Health Innovation, Murdoch Childrens Research Institute (Dr Best's comments are in bold and all pagereferences are to the document "Main document – Marked copy)

We thank Dr Best for very valuable comments and have responded to her comments in the following way:

1.

The readership of BMJ Open are not all ethicists and I suspect would be interested in the empirical bioethical approach. However, I felt there needed to be further information about this approach early on in the manuscript to help guide the reader. I was unclear of the benefits (and limitations) of this analysis.

Reviewer Dr Best rightly observed that readers of BMJ open might not be familiar with the methodological approach of empirical bioethics and would need further information. We have in accordance with this extended this section and, in more depth, explained the approach. The following is now stated in the article (p. 7-8):

The study's methodological framework is empirical bioethics,[30-32] a growing field of research.[33] Empirical bioethics is a heterogeneous field that combines empirical research – commonly qualitative empirical research – with an ethical or philosophical analysis.[32, 34] Just as other qualitative research methods, it involves a detailed analysis of descriptions and views given by interviewees on a particular subject, and a focus on complexities. However, the particular value of empirical bioethics rests with the way the qualitative analysis is combined with for example conceptual analysis and philosophical and ethical discussion.[30, 32, 34] The combination of qualitative analyses with philosophical or ethical analyses has proven to be of much value: it can refine an ethical discussion within a medical practice through its close attention to concerns that arise within this practice, without losing sight of the specific context, while ensuring that theoretical philosophical and/or ethical discussions contribute to concerns within the concrete medical practice. In this way, such combined analyses can contribute to the improvement of care. In the present study, we identify themes that include concerns held by the interviewees, engage with the results of the thematic analysis, identify norms and values, contextualize the identified themes against previous relevant analyses, and discuss the empirical findings in relation to previously identified ethical concerns and discussions (here called an empirical bioethical discussion).

2.

Internationally, much genetic carrier screening is offered in early pregnancy so this study will have missed this. In itself this is not a problem as this study focuses on PECS but there would be value in raising testing in early pregnancy to put this study in context.

We have considered this comment and find it valuable especially in order to clarify that we in this study only focuses on *preconception* expanded carrier screening and not carrier screening in early pregnancy. The Groningen pilot (being the first in the world) only offered screening to couples not yet being pregnant and this was also the case during the times for data collection. *The interviewees offering this test were experienced with PECS only and for that reason only could reflect on that.* However, the reviewers' comment is very valuable in order to pinpoint exactly what we are focussing on in the article. We have added references in order for an interested reader to get more information on ECS in early pregnancy and provided articles where both these offers (in the Netherlands) are discussed. These additions are made in the Methods-section under the title "Setting" (p. 5) formulated in the following way:

Internationally, much genetic carrier screening is offered in early pregnancy.[19-21] However, this is not the case in this study since the focus is only on a test that is taken before conception.

3.

In the introduction there is a comment that “Important qualitative research has been conducted...” could you expand on what this is and how is it important? I wondered if it was more that it was highly relevant to this study? The introduction closes with a sentence on what was done i.e., thematic analysis and a bioethical discussion. A clear aim of the study would be of help here, so the reader understands the purpose of the paper.

Thank you for this valuable comment. In the Introduction we wanted to draw attention to the only other qualitative study with GPs that has been made in the Netherlands on PECS. However, that study has a different focus on feasibility so in that sense it is not highly relevant for our study. However, as it is formulated in the article we understand that this is how it is perceived. We have now changed the wording in the Introduction to “previous qualitative research” instead of “important qualitative research” which in a better way would underline why we include that study in the Introduction. (Page 4)

We have also included a clear aim with the study as the last sentence in the Introduction. (Page 5)

4.

In the methods section I was unclear if we needed the detail about the pilot study. I found it confused this study with (I am assuming) the previous pilot work. There were no dates as to when these activities happened. The explanation about the PECS test was helpful – while the cost covers the DNA-lab test it may be interesting to readers to outline what is not covered. Are the trained GPs expected to do pre and post test counselling? Could the interview guide be added to the supplementary information?

Thank you for this valuable comment. We have looked at this part again and agree that the mentioning of a pilot-study might be confusing for the reader. However, we do find it important to include the pilot-study in order to provide context to the reader but we have made changes in order to clarify the process from pilot-study to a general offer. One example is that we only state the current number of diseases namely 70 and do not include that screening was offered for only 50 in the pilot-study. These changes are made in the Abstract, and in the Methods-section under “Setting” (p. 5).

In the Methods-section we have added the year for when the test was developed. (P.5)

Regarding costs there are no additional cost, therefore we have kept the original writing with stating the cost of 950 euro/couple. (P.5)

Regarding counselling the GPs do not perform post-test counselling, they refer the couple to clinical geneticists. This could of course be of value for the reader so we have added this information in the section “setting” on page 5.

The interview guide has been added to the supplementary information.

The results section opened directly with themes and quotes from the participants. Could there be some information about the characteristics of the participants e.g metro/rural, experience with PECS as is relevant. An outline of the themes to be presented would also be of help to signpost the reader. I found the presentation of the themes a little confusing. The second

theme was clearly presented though there were subdivisions that were then not followed up with more detail. I was unsure why the first theme 'choice and its complexities' is not contained with the second them of 'complexity of choice...'. The discussion about entangled existential genetics was of interest but felt like it belonged in the discussion section.

Characteristics: We have added characteristics concerning metro/rural in the Data collection (page 6).

Regarding experience with PECS we describe in the Data collection (page 5-6) that participants are from two groups: either those who had experience in offering PECS and those who referred patients for it.

Dr Best has indicated the need for signposting and clarification regarding the result-section. This is a valuable comment. The need for signposting is clear and we have therefore added an "introduction" to the results (p. 9) were we also underline that the thematic analysis and the bioethical discussion is "held together" in accordance with the chosen methodology. We hope this will clarify how the different parts in the result section are connected. The following is now stated in the document:

The first theme identified in the thematic analysis of interviews with GPs on PECS is “choice and its complexity”. After presenting the thematic analysis we offer an empirical bioethics discussion and argue that it highlights the need for facilitating shared relational autonomous decision-making within the couple. The second theme is “PECS as prompting existential concerns”, which includes two sub-themes: “prevention of suffering” and “the test within the framework of societal concerns”. We discuss also this theme in the context of bioethics and argue that it should preferably be understood in terms of an entangled existential genetics that brings out ethically pertinent aspects of the practice of PECS.

When we now have signposted in a clearer way 1. The themes and subthemes in the thematic analysis and 2) how the thematic analysis and the bioethical discussion is *combined* in the result we hope that these changes in a clearer way indicate why the bioethical discussion on entangled existential genetics therefore preferably should be in the Result section and not in the Discussion.

Mention is made in the limitations of the relatively small sample. I would mention the in-depth nature of the interviews as a strength of this study.

We thank Dr Best for this valuable comment. The in-depth nature was mentioned in the “strengths and limitations of the study” (p. 3) but the reviewer is right that it is omitted in the Discussion. We have included this perspective even there (p. 16).

VERSION 3 – REVIEW

REVIEWER	Best , Stephanie Australian Institute of Health Innovation, Centre for Healthcare Resilience and Implementation Science
REVIEW RETURNED	15-Nov-2021
GENERAL COMMENTS	Many thanks to the authors for revising their manuscript in line with the comments they received. I found this version flowed well and think the BMJ Open readership will find it very interesting.